# Occurrence and Exposure Assessment of Nickel in Zhejiang Province, China

**DOI:** 10.3390/toxics12030169

**Published:** 2024-02-22

**Authors:** Junde Han, Ronghua Zhang, Jun Tang, Jiang Chen, Chenyang Zheng, Dong Zhao, Jikai Wang, Hexiang Zhang, Xiaojuan Qi, Xiaoli Wu, Qin Weng, Jinping Zeng, Jiaolan Du, Min Zhang, Yinyin Wu, Biao Zhou

**Affiliations:** 1Zhejiang Provincial Center for Disease Control and Prevention, Hangzhou 310051, China; 2021112012151@stu.hznu.edu.cn (J.H.); rhzhang@cdc.zj.cn (R.Z.); jtang@cdc.zj.cn (J.T.); jchen@cdc.zj.cn (J.C.); cyzheng@cdc.zj.cn (C.Z.); dzhao@cdc.zj.cn (D.Z.); jkwang@cdc.zj.cn (J.W.); hxzhang@cdc.zj.cn (H.Z.); xjqi@cdc.zj.cn (X.Q.); xlwu@cdc.zj.cn (X.W.); 881012022119@hmc.edu.cn (Q.W.); 2Department of Epidemiology and Health Statistics, School of Public Health, Faculty of Medicine, Hangzhou Normal University, Hangzhou 311121, China; zjp18905942914@163.com (J.Z.); 2022112027009@stu.hznu.edu.cn (J.D.); zhangmin990205@163.com (M.Z.)

**Keywords:** Nickel, contamination, food safety, dietary exposure, risk assessment, Zhejiang Province

## Abstract

Nickel (Ni) is a silver-white metal with high antioxidative properties, often existing in a bivalent form in the environment. Despite being the fifth most abundant metal on Earth, anthropogenic activities, including industrial processes, have elevated Ni levels in environmental media. This study investigated Ni contamination in various food groups in Zhejiang Province, China, mainly focusing on Ni levels in beans, vegetables, aquatic foods, meat products, cereal products, and fruits. A total of 2628 samples were collected and analyzed. Beans exhibited the highest Ni content in all samples. The overall detection rate of Ni was 86.5%, with variation among food categories. For plant-origin foods, legumes had the highest Ni concentration while for animal-origin foods, shellfish showed the highest median Ni concentration. The results indicate generally acceptable Ni exposure levels among Zhejiang residents, except for children aged 0–6. Beans were identified as the primary contributor to high Ni exposure risk. The paper suggests monitoring Ni contamination in food, especially for vulnerable populations, and provides insights into exposure risks in different age groups.

## 1. Introduction

Nickel (Ni) is a highly antioxidative silver-white metal and the fifth most abundant metal on Earth, after iron, oxygen, silicon, and magnesium [1,2]. Under natural conditions, Ni in environmental media primarily originates from soil migration caused by factors such as rock weathering, soil deposition, riverbed sedimentation, and volcanic eruptions [1]. Although various valence states can exist under specific conditions, Ni compounds typically exist in the environment as Ni^2^, which has low solubility in water [3].

Ni exhibits stronger antioxidative properties than iron and is commonly used in electroplating, coatings, alloys, and industrial equipment [4]. According to a summary of US mineral commodities, global Ni reserves are distributed mainly in Australia (Australia), Canada (North America), Cuba (North America), Indonesia (Southeast Asia), and Russia (Eastern Europe) and China (East Asia) [5]. In recent years, the Ni content in soil has increased due to the acceleration of global industrialization and anthropogenic activities [6]. Human activities, including industrial wastewater generation, fossil fuel emissions, industrial production, and the use and disposal of Ni compounds and alloys in electroplating and welding, have elevated Ni levels in environmental media such as water, soil, and air [7]. Organisms inhabiting these environmental media can be affected by increased Ni levels and the accumulation of Ni content [2].

In biology, Ni predominantly exists in the form of Ni^2+^. In biology systems, all forms of Ni intake, except for Ni carbonyls, are associated with the independent action of Ni^+2^. The solubility of Ni^+2^ affects its toxicity mechanisms [8]. Owing to differences in the role of Ni in biological activities and its natural content in the environmental media where organisms live, Ni content varies among different species [9,10]. As an essential trace element in certain crops, Ni is an important constituent of the enzymes involved in reactions and its bioavailability could affected by nanoparticle exposure [11,12]. Owing to their significant Ni enrichment, these agricultural crops (e.g., beans) are used as plants with phytoremediation effects to concentrate Ni in a smaller percentage of the soil area [13]. In the soil, Ni is absorbed by crops through the roots and accumulates in edible parts such as stems, leaves, flowers, and fruits [14]. Some aquatic organisms also accumulate large amounts of Ni because of their benthic and filter-feeding characteristics [15,16]. In distinct ecosystems, the transfer of Ni between different trophic levels exhibits a biologically diluted effect in aquatic ecosystems. Organisms at higher trophic levels have lower Ni levels after ingesting organisms with high Ni contents from lower trophic levels. In contrast, terrestrial ecosystems have both biodilution and biomagnification [17].

Ni is an essential element for certain plants, but its biological function in humans remains unclear [18]. While the soluble characteristic of most Ni compounds allows them to pass into human feces, exposure to various forms of Ni can still result in a range of health problems [19]. Acute Ni exposure, often through inhalation, can lead to serious conditions [20]. There are also diverse pathways for chronic Ni exposure, with multiple modalities of contact, especially with the digestive tract [21,22]. Ni intake through drinking water and food can damage the human body, including the reproductive system and skin, and cause fatal toxicity [7,9,23]. Moreover, Ni-containing alloys, which are extensively used in food processing, can be transferred to food through contact or sterilization [24,25,26]. These findings highlight the importance of monitoring Ni contamination in food.

Severe global initiatives have been undertaken to control Ni levels in food products, but distinct results have been found between different regions. In the European market, chocolate has a relatively high Ni content, and its levels are generally comparable [27,28,29]. In the Canadian region, despite having substantial Ni reserves and economic development, no study has identified Ni pollution [30,31]. Russia faces relatively severe environmental Ni contamination, with some commercially available foods significantly exceeding the standard [32,33]. Rice locally produced in Australia has a relatively lower Ni content than those produced in Southeast Asian countries [34]. In East Asia, China has generally higher Ni content in food than in the neighboring Japan [17,35]. Overall, the type of food significantly influences the Ni content owing to the different transfer patterns in the ecosystem and characteristics of organisms [36]. Nuts, legumes, chocolate, shellfish, and similar foods have the highest Ni contents. Anthropogenic environmental Ni pollution may also be a potential source of Ni in food [8,37,38].

As such, the assessment of Ni exposure risk via dietary approaches is essential. Consequently, various international organizations have established oral Ni exposure limits. The US Environmental Protection Agency (USEPA) recommends a reference dose (RfD) of 20 μg/kg·bw per day. The European Food Safety Authority initially set a tolerable daily intake (TDI) of 2.8 μg/kg in 2015 that was later modified to 13 μg/kg·bw [8,9]. Currently, Ni exposure risk assessments employ multiple methods. The simple distribution method is a widely used model for dietary exposure assessments that analyzes different statistical quantities of consumption or concentrations of chemical pollutants. In a nationwide dietary study in Italy, a simple distribution method was employed to calculate the dietary exposure risk of Ni using the mean and P95 percentiles of consumption [28]. Although the simple distribution method was built based on point assessment, a relatively classic exposure model has been widely applied owing to its simplicity, ease of implementation, and more comprehensive results [39]. Moreover, Ni is often assessed for cumulative exposure risks, along with other heavy metals. Although some studies use the margin of exposure method [37], most adopt the target hazard quotient (THQ) method, provided by the USEPA’s Risk Assessment Guidance for Superfund, to quantify cumulative risk. THQ can also be called the hazard index (HI) [17,40]. The simple distribution and THQ methods were used in the current study [17].

Zhejiang Province has a diverse range of natural resources. The background values of Ni in the soil from Zhejiang Province are lower than the national average, indicating that the natural environment has relatively low Ni content [41,42]. Zhejiang Province has not only been a high-output area for the development of heavily polluting industries in China over the past 20 years but also one of the major regions for electronic waste dismantling in the country [43,44].The widespread presence of businesses and industries in the region has increased anthropogenic activities and the potential for Ni contamination. Zhejiang Province is a representative area for market-sold food. Previous studies explored Ni concentration in specific foods and associated health risks, providing a limited perspective; therefore, more recent updates could be beneficial [45,46,47]. This study aimed to determine Ni occurrence in multiple food groups and assess the dietary exposure risk in Zhejiang Province. Food consumption data for Zhejiang residents from 2014 to 2017, based on food safety monitoring data for chemical pollutants from 2018 to 2019, were analyzed.

## 2. Materials and Methods

### 2.1. Sampling

Food sampling followed the formula below [48]:(1)$$N=\frac{Z^{2}\times \left[P\times \left(1-P\right)\right]}{d^{2}}$$Here, *N* is the sample size, *Z* is the 95% confidence level at 1.96, *P* is the expected percentage of samples containing toxins (50%), and *d* is the precision of 10%. Using this formula, we estimated that the minimum sample size required was 96.04. However, the sample collection scope includes not only Ni but also other metals. Due to the lower concentration of p in Ni than other metals, the minimum sample size was expanded to 1824. A multistage random sampling method was employed in 11 cities across Zhejiang Province from 2018 to 2019, and 2628 food samples were collected from Zhejiang Province through various stages, including the breeding and acquisition link (field) and intermediate link (market, store, online commercial), along with cereal products (*n* = 27), beans (*n* = 5), vegetables and vegetable products (*n* = 392), aquatic food and products (*n* = 1404), meat products (*n* = 691), and fruits (*n* = 145). All samples were stored at −20 °C with cooler and dry ice and immediately transported to the laboratory under refrigeration until further experimentation. In principle, the sample quantity should meet the needs of laboratory testing and retention, be as representative as possible, objectively reflect the pollution status of the sample, and be operationally feasible for practical monitoring and sampling. Generally, the sampling quantity for each sample should not be less than 500 g (mL). For individual packages weighing more than 250 g (mL), 3–4 packages were collected for each sample. For samples weighing less than 250 g (mL), 5–8 packages were collected.

### 2.2. Chemical Solution

Nitric acid (chromatographic purity) and hydrogen peroxide (chromatographic purity) were purchased from Merck (Darmstadt, Germany). Ni standard stock solution and tuning solution (Li, Y, Ce, TI, Co, ρ = 10 ng/mL) were purchased from CNW Technologies GmbH (Düsseldorf, Germany). Water was distilled and purified using a Millipore water purification system (Millipore Ltd., Bedford, MA, USA). Samples were subjected to microwave digestion with nitric acid and selectively predigested at 120 °C on a digestion block if they were difficult to digest.

Mixed standard solution was prepared using Ni standard stock solution and diluted into standard concentration (ρ = 1.0 mg/L) with nitric acid solution (2:98). The standard work curve liquid was prepared using mixed standard solution and diluted with nitric acid solution (2:98) into standard serious reagent with concentrations of 0.0, 0.5, 1.0, 5.0, 10.0, 50.0, 100.0, and 300 ng/mL. The internal standard working solution was an internal standard solution (10 μg/mL) diluted into 1 μg/mL with the 2:98 nitric acid solution. The instrument tuning working solution was diluted using instrument tuning solution with a nitric acid solution (2:98) to reach a concentration of 1 ng/mL. All liquids were stored at −20 °C for subsequent chemical experiments.

### 2.3. Sample Analysis and Quality Control

The Ni concentration was determined by local laboratories in Zhejiang Province, according to the Manual for China National Food Contamination and Harmful Factors Risk Monitoring in 2018 [49]. Pretreatments were performed before the formal analysis. Different types of procedure samples were treated with different pretreatments based on their types before testing as follows: for high-fat, high-protein, high-starch, and high-fiber samples, after sampling and addition of nitric acid, cold digestion was required for at least 1 h; for meat and canned aquatic products, the edible parts were homogenized for sampling and then stored in a −18 °C freezer for later use.

After pretreatment, quantities of samples varied among different types and were subjected to a Mars-6 microwave digestion system (CEM, Charlotte, NC, USA). Nitric acid (6 mL to 8 mL) was added and allowed to stand for 1 h. For hard-to-digest samples, an overnight stand was performed, followed by the addition of 1 mL hydrogen peroxide. Digestion was carried out according to the optimized microwave digestion program. After complete digestion, the solution was digested to a volume of 25 mL with ultrapure water and mixed, and a blank experiment was conducted simultaneously. The microwave digestion conditions were as follows: in Step 1, the temperature was controlled at 120°, with a 5 min ramp time and 5 min isothermal time; in Step 2, the temperature was controlled at 150°, with a 5 min ramp time and 10 min isothermal time; and in Step 3, the temperature was controlled at 190°, with a 5 min ramp time and 25 min isothermal time. The resulting solution was introduced into the NexION 300D Inductively Coupled Plasma Mass Spectrometer (ICP-MS, PerkinElmer, Waltham, MA, USA) equipped with a concentric nebulizer and collision reaction cell. The sample solution was analyzed in the ICP-MS to obtain the corresponding signal response ratio. The Ni concentration in the test solution was calculated based on the standard curve.

A weighed amount of 0.5 g was used, and the volume was adjusted to 25 mL. The limit of detection (LOD) was 3.5 μg/kg for vegetables and fruits and 3.0 μg/kg for other samples. All participants received rigorous training based on the manual standards. Quality control included additional quality control samples and standard addition recovery experiments. Specifically, one quality control sample was taken for every 10 samples. Certified reference materials (CRMs) were used for quality control samples. The sample concentrations below LOD were set at 1/2 LOD [50].

### 2.4. Food Consumption Rate

Food consumption data were obtained from a dietary survey of Zhejiang Province residents conducted between 2014 and 2017. The survey areas included 10 administrative regions and was performed using the probability-proportional-to-size method. In total, 16.60%, 70.14%, and 13.26% of the survey respondents were aged 0–14, 15–59, and ≥60 years, respectively, based on the strategy of the sixth nationwide population census conducted by the National Bureau of Statistics in 2011. Three blocks were chosen randomly from each survey spot, and from each selected block, two villages were chosen. Fifty participants were selected from these villages. The participants were asked about their food habits over 12 months using a food frequency questionnaire, which covered personal information, frequency of selected food consumed per day, and food weight per consumption. Personal information including name, sex, age, and body weight was collected. Body weight was measured using a portable weight scale. The weight of food consumption was estimated using domestic measuring instruments, food models, or food graphs. Approximately 19,000 individuals were included in the food consumption survey, and they were classified into five groups according to age: 0–6, 6–13, 11–17, 18–59, and ≥60 years [51].

### 2.5. Contamination Assessment Standard

Ni levels were evaluated based on the GB 2762–2022 Limits of Heavy Metals in Food under the National Food Safety Standards, which set the overstandard Ni levels at 1 mg/kg [52].

### 2.6. Risk Assessment

The residents’ dietary exposure to Ni was analyzed using the HQ formula approved by the USEPA as follows:(2)$$\mathrm{EDI}=\frac{C\;\times \;D\;\times \;T}{BW}$$
(3)$$\mathrm{HQ}=\frac{EDI}{RfD}$$Here, EDI (i.e., estimated daily intake; μg/kg·bw·d) is the estimated daily intake of participants; *C* (mg/kg) is the concentration of Ni in certain foods; *D* (g) is the weight of food consumed per day by participants in each age group, calculated as food frequency plus food weight per consumption; *T* is the transfer rate of the selected food; *BW* (kg) is the average weight of each age group; *RfD* (μg/kg·bw·d) is the recommended reference dose; and HQ is the hazard quotient. The transfer rate was defined as 1 under the assumption that the heavy metal concentrations between food and human body ingestion were equal. *RfD* was set to 0.02 mg/kg·bw·d as recommended by the USEPA [53]. An HQ ≥ 1 was defined to indicate an unacceptable exposure risk resistance in the population.

THQ is used to estimate exposure risks in specified food categories or age groups and is counted as the summation of *HQ* using the following equation:(4)$$\mathrm{THQ}=\sum_{1}^{n}HQ_{n}$$Here, *n* is the number of specific food categories or age groups.

### 2.7. Statistical Analyses

The detection rates and overstandard rates of Ni in different foods were examined using the chi-squared test. Normality tests were used to estimate data distribution, and statistical significance was determined using either Student’s *t*-test or the Wilcoxon rank-sum test, depending on the normality of data distribution. A skewed distribution was observed in the Anderson–Darling test (*p* < 0.05) and the histogram. Hence, the Wilcoxon rank-sum test was used for analysis. The four consumption modes were as follows—Mode A: median consumption and concentration; Mode B: median consumption and the 95th percentile of concentration; Mode C: the 95th percentile of concentration and median consumption; and Mode D: the 95th percentile of concentration and the 95th percentile of consumption [54]. All statistical analyses were conducted using Excel and R software 4.2.2. A *p* value of <0.05 was considered significant.

## 3. Results

### 3.1. Sample Characteristics

A total of 2628 samples were collected. Among these, 5, 1405, 690, 365, 27, and 136 samples were beans, aquatic products, meat products, vegetables and vegetable products, cereal products, and fruits, respectively, and the median Ni concentrations were 1.740, 0.084, 0.078, 0.0701, 0.102, and 0.059 mg/kg, respectively. The occurrence of Ni is listed in Table 1. Beans exhibited the highest Ni content. The overall detection rate of Ni was 86.5% and exceeded 80% in most food categories. The detection rates for beans, cereal products, and fruits were all 100%, whereas those for vegetables and products, meat products, and aquatic food and products were 97.81%, 85.80%, and 82.28%, respectively. Notably, although the detection rate was rather high, only 5.07% of meat products, 3.56% of aquatic food and products, and 1.64% of vegetables and vegetable product samples exceeded the national standard of 1 mg/kg. In contrast, 100% of bean samples had Ni levels above the limits. Cereal products and fruits had a relatively low rate of 0.00%. The overstandard rates of Ni across various categories were low, ranging from 0% to 5.07%, except for beans (100%).

### 3.2. Plant-Origin Foods

Most plant-origin foods were reclaimed with a relatively high detection rate, approximated to 100%, and low proportion of exceeding the standard, except for beans, which scored the highest overstandard rate of 100%. For vegetables and vegetable products, legumes had the highest Ni level at 0.330, followed by pickled vegetables, tubers, other vegetables, and leafy vegetables at 0.172, 0.118, 0.053, and 0.032 mg/kg, respectively. Interestingly, pickled vegetables had higher Ni levels than other fresh vegetables except for legumes.

### 3.3. Animal-Origin Foods

For animal-origin foods, shellfish had the highest median Ni concentration, followed by canned fish, sea crustacea, gastropods, other processed aquatic foods, freshwater crustacea, freshwater fish, and sea fish. For freshwater aquatic foods, products with lower trophic levels had higher Ni concentrations than those with higher trophic levels (e.g., crustacea and shellfish). In addition, freshwater products were more abundant than their marine counterparts.

Among processed animal products, canned fish had a high Ni concentration, and the concentrations were higher than those in fresh fish. For meat products, the highest median concentrations were in sausages, followed by Chinese bacon and other meat products.

### 3.4. Current Status and Characteristics of Food Consumption

The food consumption statuses by age group are shown in Table 2. The mean total consumption levels of beans, meat products, vegetables and vegetable products, aquatic food and products, cereal products, and fruits were 0.086, 0.323, 3.826, 1.519, 0.203, and 2.489 g/kg·bw, respectively. The Wilcoxon rank-sum test indicated significant differences (*p* < 0.05) in consumption among the age groups, with fruits and vegetables being the most consumed foods.

### 3.5. Dietary Risk Assessment of Nickel by Age

The EDI and distribution of Ni-contaminated foods consumed in Zhejiang Province by age group are shown in Table 3. For consumption mode D, the EDI values for 0–6-year-old children, school children, teenagers, adults, and the elderly were 21.656, 1.682, 8.838, 9.923, and 11.4 μg/kg·bw·d, respectively. The THQ values and contribution rates of diverse foods by age groups are shown in Table 4. The highest THQ was 0–6, and only this THQ exceeded the safety threshold in high consumption and contamination conditions among the five age groups, with a value of 1.078. This indicated an unacceptable exposure risk. For other consumption modes, the Ni exposure risk was acceptable.

Beans and vegetables and vegetable products were the main sources of Ni dietary exposure for the low consumption status (modes A and C), with a total consumption of over 50%. Meanwhile, vegetables were the main sources of exposure for the high consumption status (modes B and D).

### 3.6. Daily Safe Consumption for the Population Aged 0–6 Years

Owing to the dietary exposure risk of the participants aged 0–6 years in Zhejiang Province, the safe amount of daily consumption without causing an exposure risk of Ni was calculated by P95 and the maximum Ni level (Table 5). The daily safe consumption determined by the P95 of Ni level was greater than the present study among all categories of food, especially fruits and beans, exceeding almost three times and one time, respectively.

## 4. Discussion

The Ni contents among foods could have been caused by multiple traces and were observed at different levels as per their categories [55]. The current study found that in Zhejiang Province, Ni concentrations in food were generally at lower levels; beans had the highest Ni concentration among the six food categories analyzed. The mean Ni concentrations in the current study were higher than those in a Chinese market-basket study for aquatic and cereal products, but not for beans and vegetables, with cereals, beans, vegetables, and fish having mean Ni concentrations of 0.56 mg/kg, 5.11 mg/kg, 0.09 mg/kg, and 0.02 mg/kg, respectively [17]. A study from Germany collectively analyzed legumes, nuts, oilseeds, and spices and found a mean Ni level of 1.562 mg/kg (range: 0.064 to 5.35 mg/kg) [27]. Consistent results were found in the present study. Ni contamination in food can stem from both anthropogenic and environmental sources. Studies suggest that Ni levels in the soil and riverbed sediments in Zhejiang Province are generally at a medium-to-low pollution level, being influenced to varying degrees by human activities [56,57]. Therefore, the elevated Ni contamination levels observed in the food in this study are likely attributable predominantly to anthropogenic activities.

Among plant-origin foods, the mean Ni level was the highest in legumes at 0.172 mg/kg, higher than the 0.05 mg/kg value obtained from agriculture bases in Zhejiang Province [41]. Ni is an essential structural component of urease and hydrogenase in legume plants, and Fabaceae plants (legumes, beans) have the same characteristics of urease formation through purine catabolism and the accumulation of urea as a major reserve form for translocating nitrogen from roots into shoots [11,57]. Hence, the median concentration in tubers is higher than that in leafy vegetables. In a similar study in Jiangxi Province, China, the median Ni concentration in tubers was 0.10 mg/kg whereas that of leafy vegetables was 0.061 mg/kg [58]. In the second French Total Diet Study (TDS), the mean Ni level was higher in root vegetables than in other vegetables (0.105 mg/kg vs. 0.093 mg/kg) [29]. However, some studies have reported conflicting findings wherein leafy vegetables have a greater ability to accumulate Ni than do root vegetables, although the specific reason for this observation requires further investigation [41]. Cereal products and fruits in this study had relatively lower Ni levels than other vegetables. The mean Ni concentration was approximately 0.140 mg/kg, consistent with the mean Ni concentration in flavor products produced in China [59]. In an Italian study, the mean Ni content in bread was 0.121 mg/kg, higher than that in the present study at 0.102 mg/kg [28]. Meanwhile, the Ni concentrations in fruits in the present study (0.059 mg/kg) were lower than those in the second French TDS (1.12 mg/kg) [29]. Ni has been proven to be transferred to processed foods during their production [26]. Therefore, picked vegetables may contain higher Ni concentrations than fresh vegetables. The mean Ni concentration in soybeans in the current study (9.21 mg/kg) was lower than that in a study in Luxembourg (3.00 mg/kg) [60].

Meanwhile, for animal-origin foods, the mean Ni concentrations in aquatic foods and products (0.261 mg/kg) were higher than that obtained from Shaoxing, Zhejiang Province, China but lower than that in France (0.041 mg/kg and 0.299 mg/kg, respectively) [61,62]. Shellfish are the most abundant low-trophic-level aquatic food. In Italy, crustacea and mollusks had a higher mean Ni concentration of 0.626 mg/kg than fish (0.046 mg/kg) [28]. Processed shellfish are more likely to be exposed to Ni and be contaminated because of benthic and filter feeding [63]. Biological dilution was observed in the transfer of Ni among aquatic foods, which could mean higher Ni concentrations in low-trophic-level aquatic foods than in high-trophic-level aquatic foods [17]. In addition, Ni was detected in all aquatic food categories, and the Ni concentration was higher in freshwater products than in marine products. The Ni content is significantly affected by the dilution effect [64]. A previous study reported that the Ni concentration decreases from the estuary to the inner bay under the effect of river input [65]. Thus, aquatic ecosystems can be affected by the contrast between marine and freshwater environments, resulting in conflicting data. In a survey of potentially toxic elements in canned foods in Hangzhou, Zhejiang Province, China, canned fish had a mean Ni concentration of 0.265 mg/kg, higher than the 0.647 mg/kg detected in our study [66]. Similar to pickled vegetables, processed foods (e.g., canned fish or processed meat) have higher levels of Ni than fresh foods. Ni is transformed via food contact materials and is likely to be transferred to food during processing in the factory [67]. However, a distinct result was found in a German study that showed a lower level of Ni accumulation in canned tuna (0.175 mg/kg) than in fresh tuna (0.374 mg/kg) [27]. This value was also lower than that in canned fish in the present study. The mean Ni level in sausages was 0.262 mg/kg, lower than that in fresh meat collected in Zhejiang in 2016 (0.055 mg/kg) and sausage products sampled from an Armenian market (1.55 mg/kg) [39,48].

In general, Ni exposure was at acceptable levels among the residents of Zhejiang Province, except for the population aged 0–6 years, in whom the EDI was the highest and consumption of Ni-contaminated foods exceeded the recommended TDI value of 13 μg/kg·bw [8]. EDI values for consumption modes C and D in adults in the current study exceeded the 3.9 μg/kg·bw·d that was derived from the south coast in China [17]. In consumption modes A and B, the EDI values were 0.404 μg/kg·bw·d and 1.181 μg/kg·bw·d for adults and 0.833 μg/kg·bw·d and 2.227 μg/kg·bw·d for children aged 0–6, lower than those in French adults and children at 3.76 μg/kg·bw·d and 7.44 μg/kg·bw·d and at 3.83 μg/kg·bw·d and 7.44, respectively [29]. The above EDI results indicate that the younger-age population have a higher exposure owing to their body weight. However, with the underdevelopment of the body, the health effects from heavy metal exposure may be more severe. Therefore, heavy metal content from dietary exposure should be more closely monitored in younger children. Beans were the primary contributors to the exposure risk to high Ni levels. In a chronic dietary exposure monitoring of the Belgian population, beans contributed to a high Ni exposure in three diverse age groups, namely children (3–9 years), teenagers (11–17 years), and adults (18–64 years), with rates of 26%, 19%, and 16%, respectively [68]. Collectively, these findings and those in the current study support that elevated Ni levels may be a result of high Ni levels in food. Moreover, drinking water serves as another crucial pathway for the daily intake of exogenous substances. To complement the results of the present study, we obtained drinking water consumption and contamination data from publications to calculate the exposure risk using the same approach as mentioned above. The risk value for Ni exposure was calculated to be 0.001 under high-consumption conditions, significantly lower than 1, which indicated an acceptable level [69,70].

A TDS study in China showed that the southern coast area had an HI value of 0.2, higher than the THQ results of consumption modes A and B from all populations in the present study, indicating that average consumption contributed to an acceptable health risk [17]. To measure the reference food intake to avoid the risk of Ni exposure, we used the conservative method to analyze the amount of safe daily consumption according to the P95 level of Ni for the population aged 0–6 years. The results indicated that the most commonly consumed foods, in declining order, were beans, meat products, vegetables and products, aquatic food and products, cereal products, and fruits. Unavailable amounts of recommended consumption were found in among all categories of foods, especially for beans (1296.041 g) and fruits (1257.417 g). The consumption levels of beans and fruits were a 100-fold and 3-fold higher than the recommended limits, respectively.

The present study had the following limitations. Cereals are a primary dietary source among the Chinese [71]. Only a limited variety of cereals (rice, flavor products) were sampled in the current study, including bread, cookies, and egg rolls. This may have led to an underestimation of the risk of Ni exposure from cereal consumption. Animal fat was the only food category listed on the GB 2762–2022 national standard. However, although beans had the highest overstandard ratio in the present study, they have lower Ni accumulation ability than animal fat [13,71]. Hence, the overstandard ratio could have been overestimated. To improve the precision and pertinence of the present monitoring method, it is necessary to classify the limited Ni values among diverse foods. Finally, a probability assessment is required when the risk exposure values approach or exceed the Allowable Daily Intake, as actual dietary exposures are complex, and both point assessment and simple distribution are relatively conservative in describing the distribution of dietary exposure [72]. Owing to technical limitations, this method was not used in the present study.

## 5. Conclusions

Market-sold foods in Zhejiang Province had low Ni contamination although Ni was detected in most food samples. Athropic activities could be a potential contamination resource. Beans had the highest Ni concentration, with sustained high levels of overstandard rates; other food categories had low Ni concentrations. Among plant-origin foods, soybeans and legumes had higher concentrations of Ni. Tubers also had higher Ni levels than leafy vegetables, the reason for which requires further investigation. Among animal-origin foods, shellfish had the highest Ni concentration owing to them being inherently low-trophic-level aquatic food. Ni concentration was higher among low-trophic-level foods than among high-trophic-level foods, revealing a biodilution effect. For aquatic foods, Ni concentrations differ between those from marine and from freshwater environmental media. This may be because of the dilution of Ni from freshwater to the ocean. Further, processed foods such as pickled vegetables and canned fish had higher Ni levels owing to the transfer of Ni during processing. In summary, this exposure risk assessment study found an acceptable risk of Ni exposure in the population of Zhejiang Province, including in children aged 0–6. Under extreme conditions of high exposure and contamination, children aged 0–6 have a risk value of approximately 1, which could be explained by their low body weight. Therefore, a comprehensive study in younger children is crucial.

## Figures and Tables

**Table 1 toxics-12-00169-t001:** Ni concentrations (mg/kg) in food samples collected from Zhejiang Province.

Food Categories	Food Names	Range	Mean	P50	P95	Total Number	Detection Rate (%)	Overstandard Rate (%)
Beans	Soybean	9.21–9.21	9.21	9.21	9.21	1	100	100
	Mung bean	1.22–1.38	1.3	1.3	1.372	2	100	100
	Ormosia	1.74–1.92	1.83	1.83	1.911	2	100	100
	Total	1.22–9.21	3.094	1.74	7.752	5	100	100
Meat and products	Sausages	0.002–3.38	0.262	0.095	0.978	316	91.14	4.11
	Chinese bacon	0.002–3	0.252	0.053	1.093	296	78.04	6.42
	Other meat product	0.010–0.447	0.116	0.049	0.37	7	85.71	0
	Total	0.002–4.99	0.258	0.084	1.006	690	85.8	5.07
Vegetables and vegetable products	Legume	0.161–3.35	0.813	0.33	2.61	9	100	22.22
	Tubers	0.001–3.19	0.215	0.118	0.751	240	97.08	1.67
	Pickled vegetables	0.074–0.251	0.167	0.172	0.247	4	100	0
	Leaf vegetables	0.007–0.99	0.07	0.032	0.231	78	100	0
	Other vegetables	0.002–0.365	0.07	0.053	0.169	34	97.06	0
	Total	0.001–3.35	0.185	0.078	0.639	365	97.81	1.64
Aquatic food and products	Sea fish	0.001–1.34	0.057	0.007	0.076	30	43.33	3.33
	Gastropods	0.05–0.35	0.194	0.183	0.333	3	100	0
	Other processed aquatic products	0.002–4.2	0.209	0.082	0.891	399	89.72	4.26
	Sea crustacea	0.006–0.84	0.195	0.19	0.419	150	99.33	0
	Freshwater crustacea	0.002–48.9	0.281	0.047	0.582	546	78.75	2.38
	Freshwater fish	0.002–10.5	0.149	0.01	0.422	145	53.1	1.38
	Caned fish	0.002–19.1	0.647	0.204	1.539	82	92.68	13.41
	Shellfish	0.056–1.53	0.484	0.308	1.279	50	100	12
	Total	0.001–48.9	0.261	0.071	0.852	1405	82.28	3.56
Cereal products	Total	0.025–0.696	0.136	0.102	0.302	27	100	0
Fruits	Total	0.004–0.75	0.097	0.059	0.293	136	100	0

**Table 2 toxics-12-00169-t002:** Daily ingestion rate of food (g) and body weight (kg) in Zhejiang Province.

Age Groups	Food Categories	Mean	P50	P95	Max	Age Groups	Food Categories	Mean	P50	P95	Max
0–6 Years	Aquatic food and products	29.282	0.822	17.143	200	18–59 Years	Aquatic food and products	84.612	1.333	28.571	1600
	Meat products	8.76	1	13.333	150		Meat products	17.881	1.370	15.000	750
	Cereal products	10.15	1.096	19.286	210		Cereal products	10.094	1.000	16.667	500
	Vegetables and vegetable products	101.801	3.333	40	650		Vegetables and vegetable products	227.196	6.667	64.286	1150
	Fruits	115.636	3.333	45	400		Fruits	148.755	4.658	57.143	1160
	Beans	2.795	0.667	6.767	15		Beans	4.937	1.000	10.200	171.429
	Body weight	18.988					Body weight	60.170			
7–10 Years	Aquatic food and products	42.777	0.986	21.429	300	≥60 Years	Aquatic food and products	48.751	1	28.571	1157.143
	Meat products	13.271	1.333	15	260		Meat products	10.343	1.096	14.286	300
	Cereal products	11.489	1.37	20	128.571		Cereal products	6.688	0.959	16.000	120
	Vegetables and vegetable products	153.726	5.333	50	475.714		Vegetables and vegetable products	239.424	6.667	68.571	600
	Fruits	146.753	4.11	57.143	800		Fruits	123.885	3.833	57.143	400
	Beans	4.118	0.833	8.571	68.571		Beans	5.624	1.167	14.286	100
	Body weight	35.233					Body weight	52.652			
11–17 Years	Aquatic food and products	45.727	1.027	21.836	550	All ages	Aquatic food and products	86.455	1.143	28.571	1600
	Meat products	14.831	1.667	16.029	300		Meat products	18.402	1.342	14.286	750
	Cereal products	11.958	1.37	21.429	350		Cereal products	11.544	1.014	17.143	500
	Vegetables and vegetable products	187.721	6	60	285.714		Vegetables and vegetable products	217.732	6.667	64	1150
	Fruits	137.431	4.603	51.429	360		Fruits	141.666	4.167	57.143	1160
	Beans	3.887	1	8.080	34.286		Beans	4.887	1.000	10.667	171.429
	Body weight	61.639					Body weight	56.911			

**Table 3 toxics-12-00169-t003:** Estimated daily intake of nickel among different populations in Zhejiang.

	Daily Dietary Ni Intake (μg/kg·bw·d) by Age Group																			
	0–6 Years				7–10 Years				11–17 Years				18–59 Years				≥60 Years			
Food Category	Consumption Mode																			
	A	B	C	D	A	B	C	D	A	B	C	D	A	B	C	D	A	B	C	D
Beans	0.368	0.373	4.092	4.146	0.224	0.228	1.417	1.463	0.183	0.186	0.774	0.798	0.156	0.159	1.494	1.519	0.187	0.19	3.498	3.527
Meat products	0.019	0.18	0.189	1.875	0.01	0.105	0.116	1.18	0.009	0.089	0.07	0.738	0.006	0.058	0.221	1.296	0.007	0.066	0.058	0.573
Vegetables and vegetable products	0.232	0.953	1.954	8.214	0.198	0.819	1.364	5.665	0.175	0.728	1.009	4.133	0.16	0.669	1.017	4.178	0.168	0.699	1.091	4.287
Aquatic food and products	0.082	0.299	0.912	3.878	0.05	0.192	0.563	2.211	0.035	0.141	0.383	1.520	0.034	0.14	0.406	1.726	0.03	0.126	0.44	1.806
Cereal products	0.04	0.064	0.540	0.81	0.028	0.045	0.309	0.488	0.019	0.03	0.231	0.357	0.014	0.021	0.148	0.238	0.014	0.021	0.168	0.253
Fruit	0.091	0.357	0.659	2.641	0.063	0.248	0.424	1.675	0.042	0.162	0.273	1.094	0.034	0.133	0.241	0.966	0.029	0.114	0.24	0.954
Total	0.833	2.227	8.346	21.57	0.572	1.637	4.192	12.680	0.462	1.336	2.739	8.638	0.404	1.181	3.527	9.923	0.436	1.216	5.496	11.4

**Table 4 toxics-12-00169-t004:** Exposure risk of Ni and contribution rate (%) by age group in Zhejiang Province.

Age Group	Food Category	THQ of Different Consuming Modes (Contribution Proportion, %)			
		Consumption Mode A	Consumption Mode B	Consumption Mode C	Consumption Mode D
0–6 Years	Beans	0.018 (0.442)	0.019 (0.168)	0.205 (0.49)	0.207 (0.192)
	Meat products	0.001 (0.023)	0.009 (0.081)	0.009 (0.023)	0.094 (0.087)
	Aquatic food and products	0.004 (0.098)	0.015 (0.134)	0.046 (0.109)	0.194 (0.18)
	Vegetables and vegetable products	0.012 (0.279)	0.048 (0.428)	0.098 (0.234)	0.411 (0.381)
	Cereal products	0.002 (0.048)	0.003 (0.029)	0.027 (0.065)	0.041 (0.038)
	Fruit	0.005 (0.11)	0.018 (0.16)	0.033 (0.079)	0.132 (0.122)
	THQ	0.042 (1)	0.111 (1)	0.417 (1)	1.078 (1)
7–10 Years	Beans	0.011 (0.391)	0.011 (0.139)	0.071 (0.338)	0.073 (0.115)
	Meat products	0 (0.017)	0.005 (0.064)	0.006 (0.028)	0.059 (0.093)
	Aquatic food and products	0.002 (0.087)	0.01 (0.117)	0.028 (0.134)	0.111 (0.174)
	Vegetables and vegetable products	0.01 (0.346)	0.041 (0.501)	0.068 (0.325)	0.283 (0.447)
	Cereal products	0.001 (0.049)	0.002 (0.027)	0.015 (0.074)	0.024 (0.039)
	Fruit	0.003 (0.11)	0.012 (0.151)	0.021 (0.101)	0.084 (0.132)
	THQ	0.029 (1)	0.082 (1)	0.21 (1)	0.634 (1)
11–17 Years	Beans	0.009 (0.396)	0.009 (0.139)	0.039 (0.283)	0.04 (0.092)
	Meat products	0 (0.019)	0.004 (0.066)	0.004 (0.026)	0.037 (0.085)
	Aquatic food and products	0.002 (0.076)	0.007 (0.106)	0.019 (0.14)	0.076 (0.176)
	Vegetables and vegetable products	0.009 (0.378)	0.036 (0.545)	0.05 (0.368)	0.207 (0.478)
	Cereal products	0.001 (0.041)	0.001 (0.022)	0.012 (0.084)	0.018 (0.041)
	Fruit	0.002 (0.091)	0.008 (0.121)	0.014 (0.1)	0.055 (0.127)
	THQ	0.023 (1)	0.067 (1 )	0.137 (1)	0.432 (1)
18–59 Years	Beans	0.008 (0.387)	0.008 (0.135)	0.075 (0.424)	0.076 (0.153)
	Meat products	0 (0.015)	0.003 (0.049)	0.011 (0.063)	0.065 (0.131)
	Aquatic food and products	0.002 (0.084)	0.007 (0.119)	0.02 (0.115)	0.086 (0.174)
	Vegetables and vegetable products	0.008 (0.395)	0.033 (0.567)	0.051 (0.288)	0.209 (0.421)
	Cereal products	0.001 (0.034)	0.001 (0.018)	0.007 (0.042)	0.012 (0.024)
	Fruit	0.002 (0.084)	0.007 (0.113)	0.012 (0.068)	0.048 (0.097)
	THQ	0.02 (1)	0.059 (1)	0.176 (1)	0.496 (1)
≥60 Years	Beans	0.009 (0.429)	0.01 (0.156)	0.175 (0.636)	0.176 (0.309)
	Meat products	0 (0.017)	0.003 (0.054)	0.003 (0.011)	0.029 (0.05)
	Aquatic food and products	0.002 (0.069)	0.006 (0.104)	0.022 (0.08)	0.09 (0.158)
	Vegetables and vegetable products	0.008 (0.386)	0.035 (0.575)	0.055 (0.199)	0.214 (0.376)
	Cereal products	0.001 (0.033)	0.001 (0.017)	0.008 (0.031)	0.013 (0.022)
	Fruit	0.001 (0.066)	0.006 (0.093)	0.012 (0.044)	0.048 (0.084)
	THQ	0.022 (1)	0.061 (1)	0.275 (1)	0.57 (1)

**Table 5 toxics-12-00169-t005:** Daily safe consumption (g) for the 0–6-years population in consumption mode D.

Age Group	Food Categories	Daily Safe Consumption
0–6 Years	Aquatic food and products	48.986
	Meat products	377.475
	Cereal products	445.704
	Vegetables and vegetable products	594.272
	Fruits	1257.417
	Beans	1296.041

## Data Availability

The data presented in this study are available upon request from the corresponding author due to restrictions, e.g., privacy or ethical.

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
