# Peer review of "Occurrence and Exposure Assessment of Nickel in Zhejiang Province, China"

_toxics, 2024, doi:10.3390/toxics12030169_

Round 1

Reviewer 1 Report

Comments and Suggestions for Authors

This study calculated the risk of nickel exposure in the Zheziang region. If nickel is an essential element, a certain amount is needed in the body. Is it possible to distinguish between the amount naturally present in the body and the amount caused by external exposure? Were there any bioaccumulation in food groups from the soil? Were the nickel concentration in the food in the region exceeded the guidelines? It would be interesting to read those contents as a reader. 

Line33: The meaning of the sentence is strange

Line58: Please describe the examples of “these agricultural crops”

Line119: There is no Ref46.

Line120-123: According to the equation, the minimum sample number is 96.4, so why were 2628 samples collected? And what is the basis for deciding p to be 0.5?

line138: How much did you add?

Line140: Provide the information of chemicals and MS.

Line143-154: Please add the QA/QC.

line154: There is no Ref47.

line170: There is no Ref48.

line174: There is no Ref49.

line185: There is no Ref50.

Line196-198: Please provide the explanation of the Consumption mode A-D.

Line198-199: What does it mean that “median and p95 represent different situations”?

Table1: express unit.

Comments on the Quality of English Language

There were several sentences with strange meanings. English editing is necessary.

Author Response

Comments 1: This study calculated the risk of nickel exposure in the Zhejiang region. If nickel is an essential element, a certain amount is needed in the body. Is it possible to distinguish between the amount naturally present in the body and the amount caused by external exposure?
Response 1: Thank you for your thoughtful consideration. To the best of our knowledge, nickel (Ni) has been regarded as an essential element in certain studies. However, other studies assert that nickel has no clearly identified biological function in human [1,2]. Ni is not deemed an essential element according to the Chinese dietary reference intakes—Part 3: Trace element. Our study primarily concentrates on dietary exposure rather than nutrition. Therefore, external exposure through the diet can be distinguished from naturally occurring sources. We appreciate your valuable insights.
1.     Cabrera-Vique, C.; Mesías, M.; Bouzas, P.R. Nickel Levels in Convenience and Fast Foods: In Vitro Study of the Dialyzable Fraction. Sci Total Environ 2011, 409, 1584–1588, doi:10.1016/j.scitotenv.2010.12.035.
2.     Chain (CONTAM), E.P. on C. in the F. Scientific Opinion on the Risks to Public Health Related to the Presence of Nickel in Food and Drinking Water. EFSA Journal 2015, 13, 4002, doi:10.2903/j.efsa.2015.4002.

Comments 2: Were there any bioaccumulation in food groups from the soil? Were the nickel concentration in the food in the region exceeded the guidelines? It would be interesting to read those contents as a reader
Response 2: Thank you for your recommendation. For your information, there were bioaccumulation exist in food groups from soil and mentioned in the lines 54-59. Besides, inspiring by your recommendation, we have added contents about soil in study region, in introduction section, lines 120-122 and discussion section, lines 353-358.
Though the EU legislation currently have no maximum levels for Ni in food, we used the GB 2762-2022 Chinese national standard and reported that were not exceeded the guidelines except of beans. Considering your suggestions, we have added few contents in result section in the lines 254-256.

Comments 3: Line33: The meaning of the sentence is strange
Response 3: Sorry about the original expression. The sentence is revised in the lines 36-38.

Comments 4: Line58: Please describe the examples of “these agricultural crops”
Response 5: The examples are added in the article using with “e.g.”.

Comments 5: Line119: There is no Ref46.
Response 5: The missing reference has been added in the line 113.

Comments 6: Line120-123: According to the equation, the minimum sample number is 96.4, so why were 2628 samples collected? And what is the basis for deciding p to be 0.5?
Response 6: Thank you for the information. It's important to note that the sample collection in this study extends beyond nickel to encompass other metals. Consequently, the minimum sample number was initially calculated as 96.4, considering a detection rate (p) of 0.5 and precision (d) of 0.1 (0.2*p). Given the prevalence of 0.05 in nickel, the sample size needed to be extended to 1824. The current sample size adequately covers the minimum detection rate of 0.05, rendering this data inconsequential to the overall results. However, for clarity, we have revised the expression and included the aforementioned information in lines 126-130. Once again, thank you for your reminder.

Comments 7: How much did you add?
Response 7: 1mL hydrogen peroxide was added. The number is added in line 172.

Comments 8: Line140: Provide the information of chemicals and MS.
Response 8: Thank you for your attention. Information regarding the chemicals can be found in lines 144-159. Specifically, details about the Mass Spectrometer (MS) are provided in lines 180-182. Further elaboration is provided below:

The Mass Spectrometer used is the NexION 300D ICP-MS (PerkinElmer, USA), which is equipped with a concentric nebulizer and collision reaction cell. The RF power was set at 1350W. The plasma gas flowed at a rate of 18.0 mL/min, while the oxygen flow rate was 0.065 mL/min. The auxiliary gas had a flow rate of 1.2 mL/min, and Cell Gas B flowed at a rate of 1.4 mL/min. The RPQ value was 0.45.

Comments 9: Line143-154: Please add the QA/QC.
Response 9: The specific details of quality control (QC) have added in line 188-189. If there is any information you need to know, please feel free to contact.

Comments 10: line154: There is no Ref47.
line170: There is no Ref48.
line174: There is no Ref49.
line185: There is no Ref50.
Response 10: Thank you for your reminder, all the missing reference are already added.

Comments 11: Line196-198: Please provide the explanation of the Consumption mode A-D.
Response 11: The explanation of the consumption mode A-D is provided in line 233-237.

Comments 12: Line198-199: What does it mean that “median and p95 represent different situations
Response 12: The meaning is provided above in lines 233-237, thank you for your reminder, the original expression is confused and distractive. We have revised it.

Comments 13: Table1: express unit.
Response 13: The unit is mg/kg and added in the title of table 1 in line 239. 

Reviewer 2 Report

Comments and Suggestions for Authors

Line-41-45” According to a summary 41 of US mineral commodities, over 100,000,000 tons of global Ni reserves are distributed mainly in Australia (Australia), Canada (North America), Cuba (South America), Indone- asia (Southeast Asia), and Russia (Eastern Europe) and 2,100,000 tons in China (East Asia)” these numbers sound very questionable. I suggest to remove

The validity of Equation 1, which was referenced from source 46, appears questionable, especially considering that the article incorporates a total of 45 references. It is essential to reevaluate the accuracy and appropriateness of this equation in the context of the study. Additionally, the introduction section exhibits a scattered structure and would benefit from greater conciseness and focus.

I recommend authors benefit from the relevant publications on toxicity of metals:

Application of zinc oxide nanoparticles to promote remediation of nickel by Sorghum bicolor: Metal ecotoxic potency and plant response

Environmental exposure and nanotoxicity of titanium dioxide nanoparticles in irrigation water with the flavonoid luteolin

Comments on the Quality of English Language

English is at decent level.

Author Response

Comments 1: Line-41-45” According to a summary

41 of US mineral commodities, over 100,000,000 tons of global Ni reserves are distributed mainly in Australia (Australia), Canada (North America), Cuba (South America), Indone- asia (Southeast Asia), and Russia (Eastern Europe) and 2,100,000 tons in China (East Asia)” these numbers sound very questionable. I suggest to remove

Response 1: Thank you for bringing up the concern. The inclusion of data on nickel reserves indeed raised potential confusion among readers, diverting their attention. In accordance with your request, this information has been removed.

Comments 2: The validity of Equation 1, which was referenced from source 46, appears questionable, especially considering that the article incorporates a total of 45 references. It is essential to reevaluate the accuracy and appropriateness of this equation in the context of the study.

Response 2: The concern you raised is significant. Reference source 46 does exist, and Equation 1 was indeed derived from it. The absence of references in the manuscript resulted from a formatting error in the template, which regrettably went unnoticed until later. We have conducted a comprehensive review and rectified the omission by including all the missing references. Thank you for bringing this to our attention.

Comments 3: Additionally, the introduction section exhibits a scattered structure and would benefit from greater conciseness and focus.

Response 3: It’s crucial for introduction to have a solid structure, your recommendation is helpful and I rewrote the introduction section mostly in lines 32-38, 39-40, 50-52, 66-67. Lines 108-113 is added for specific introduction of study region. I hope the current modifications meet your satisfaction.

Comments 4: I recommend authors benefit from the relevant publications on toxicity of metals:

Application of zinc oxide nanoparticles to promote remediation of nickel by Sorghum bicolor: Metal ecotoxic potency and plant response

Environmental exposure and nanotoxicity of titanium dioxide nanoparticles in irrigation water with the flavonoid luteolin

Response 4: The publications you mentioned above have been inspiring and have significantly contributed to my understanding of metal toxicity. I have thoroughly reviewed the two articles and have decided to include the first one in line 56. Thank you for providing this valuable information.

Reviewer 3 Report

Comments and Suggestions for Authors

The manuscript "Occurrence and exposure assessment of nickel in Zhejiang Province, China" by Han et al., could be an article interesting for the readers of Toxic. but in the present form, it is not acceptable for the publication.The authors' main objective is to evaluate nickel concentrations in different types of food in Zhejiang province. Some considerations:

The authors assert that the province of Zhejiang is an anthropized area, therefore the objective also includes, I imagine, the discrimination of a Ni of geogenic origin from the anthropic one, but I have not read anything in this regard.

The food samples analyzed are purchased, I imagine that the authors are certain that the products purchased are grown, fished, etc. within Zhejiang province.

Therefore it would certainly be interesting to have at least an idea of the nickel concentrations in the soil, water, and air of the study area. This would certainly make the results obtained more interesting.

In the materials and methods paragraph, I believe it is useful to describe the study area from a geological point of view, the main anthropic activities present, and their influence from an environmental and epidemiological point of view.

I recommend that the authors combine subsections 2.2 and 2.3.

I also recommend adding more accurate information on the mineralization of the samples, I believe that mineralizing a fish is certainly different from mineralizing a leafy vegetable, therefore I recommend inserting the treatment followed specifically for each type of type analyzed.

The authors write lines 140-141 "The resulting solution was introduced into the mass spectrometer using an ion collection system, and quantitative analysis was conducted using an external standard method" the authors are invited to specify which mass spectrometer they used, the reference materials used for the various types of food, the calculation of the LOD, the calculation of the recovery.

On the basis of what is recommended, evaluation of the study area, and the inclusion of data on the quality of water, soil, and air can provide further food for thought, therefore I believe that the results and discussions paragraph should be revised, the same for the conclusions.

Furthermore, the authors could distinguish natural Ni from that coming from anthropic activities that persist in the investigation area.

Author Response

Comments 1: The authors assert that the province of Zhejiang is an anthropized area, therefore the objective also includes, I imagine, the discrimination of a Ni of geogenic origin from the anthropic one, but I have not read anything in this regard.
Response 1: Thank you for your point mentioned above, the health risk assessment is both event from environment and anthropogenic. Though anthropogenic is focused in the present study, but it is also crucial to indicated the role of environment. As much as we know, there are publications noted that anthropic in Zhejiang province have more contribution to Ni than the geogenic aspect. The specifics are added in introduction section, in lines 108-113, and discussion section, in lines 318-321. Thanks again for your consideration.

Comments 2: The food samples analyzed are purchased, I imagine that the authors are certain that the products purchased are grown, fished, etc. within Zhejiang province. 
Therefore it would certainly be interesting to have at least an idea of the nickel concentrations in the soil, water, and air of the study area. This would certainly make the results obtained more interesting.
Response 2: The foods were purchased in Zhejiang province, shown in lines 131-132. Authors are certain that most of products are origin from the Zhejiang province. Besides, this study is mainly focused on dietary exposure instead of environment, but the idea of the nickel concentrations in the soil, water, and air of the study area is inspiring, thank you for mentioned. Instead of the precise concentration level, the soil background value and drinking water concentration of Zhejiang Province is referenced in lines 108-110 and lines 392-396, respectively. I hope this could meet your satisfaction.

Comments 3: In the materials and methods paragraph, I believe it is useful to describe the study area from a geological point of view, the main anthropic activities present, and their influence from an environmental and epidemiological point of view.
Response 3: It is a great idea and really useful. The description from the study area of the geological view, the main anthropic activities present and their influence from an environmental and epidemiological point of view are added in the article. In consideration of manuscript structure, I write these contents in introduction section, lines 108-113.

Comments 4: I recommend that the authors combine subsections 2.2 and 2.3.
Response 4: Thank you for your recommendation. The subsection 2.2 and 2.3 is combined into subsection 2.3 Sample Analysis and Quality Control, as shown in the line 162-191. Notably, the subsection 2.2 Chemical solution was newly added to fill in more details about experimental solution. Hope this could meet your satisfaction.

Comments 5: I also recommend adding more accurate information on the mineralition of the samples, I believe that mineralizing a fish is certainly different from mineralizing a leafy vegetable, therefore I recommend inserting the treatment followed specifically for each type of type analyzed.
Response 5: Thank you for your recommending, an accurate information on the samples mineralizing is crucial to the credibility and accuracy of research. As you mentioned above, different types of food have distinct pretreatment procedure. The details are write in lines 164-169. If there is any information required, please let me know.

Comments 6: The authors write lines 140-141 "The resulting solution was introduced into the mass spectrometer using an ion collection system, and quantitative analysis was conducted using an external standard method" the authors are invited to specify which mass spectrometer they used, the reference materials used for the various types of food, the calculation of the LOD, the calculation of the recovery. 
Response 6: Specific details of experimentation could raise reliability of the study, I’m glad to see your recommendation, and thank you for mentioned. For your question, we used the NexION 300D Inductively Coupled Plasma Mass Spectrometer (ICP-MS, PerkinElmer, USA) for quantitative analysis. The reference materials were used for quality control samples. Details of LOD calculation are also mentioned in subsection 2.3, lines 162-192 along with the others. Owing to the limitation of article, various of reference materials and the calculation of recovery is not listed. Authors are glad to provide further details if it is necessary.

Comments 7: On the basis of what is recommended, evaluation of the study area, and the inclusion of data on the quality of water, soil, and air can provide further food for thought, therefore I believe that the results and discussions paragraph should be revised, the same for the conclusions.
Response 7: Your recommendation about results, discussion, and conclusion paragraph have provided significant assistance in the writing of the paper. As you recommended, the evaluation of the study area is written in introduction sections, lines 108-113; the quality of environmental medias could be found in discussion section, lines 355-360; Besides, Ni content in drinking water is mentioned and calculated in lines 426-432, by assessing its exposure risk. The conclusion is been revised as well, in line 460.

Comments 8: Furthermore, the authors could distinguish natural Ni from that coming from anthropic activities that persist in the investigation area.
Response 8: Thank you for mentioned that. Ni concentration could be influenced by both environment and anthropic activities. Though we didn’t performed the relevant survey to distinguish where Ni coming from, we have found articles that could support the natural Ni from Zhejiang Province is lower than the national average, and could possibly affected by elevated anthropic activities. Such information could found in lines 108-113.

Reviewer 4 Report

Comments and Suggestions for Authors

Author Response

Comments 1: The reliability of exposure risk assessment of a certain population when only one metal and one route enter to body are considered is a little problematic.
Response 1: Thank you for your consideration; it has significantly enhanced the manuscript. Regarding your query, our study primarily concentrates on dietary exposure, with other routes of entry (e.g., air, contact) not being the main focus. Nevertheless, we recognize the significance of drinking water as a dietary exposure pathway. Therefore, we have gathered data from published articles and computed the Target Hazard Quotient values for drinking water in Zhejiang Province.

Comments 2: Line 43: Please replace South America with North America What kind of analytical procedure for the determination of nickel was used? The standard method or the in-house method was used for the determination of nickel? Please, provide references
Response 2: Thank you for your recommendation. The term "Sorth America" has been corrected to "North America." Regarding the analytical procedure, we have revised and provided additional details in lines 180-182. Both the standard and in-house methods have been rewritten with specific details included. This enhancement in experimental details contributes to the study's reliability, and we appreciate your suggestion. You can find the standard and in-house methods elaborated in lines 162-192. If further details are needed, the authors are pleased to provide them upon request.

Comments 3: Line 140: „The resulting solution was introduced into the mass spectrometer” Indeed? please provide the type of this mass spectrometer
Response 3: For your question, we used the NexION 300D Inductively Coupled Plasma Mass Spectrometer (ICP-MS, PerkinElmer, USA) for quantitative analysis. More details are provided below: NexION 300D ICP-MS (PerkinElmer, USA) equipped with a concentric nebulizer and collision reaction cell. The RF power was 1350W. The flow rate of the plasma gas was 18.0 mL/min, while the flow rate of oxygen was 0.065 mL/min. The auxiliary gas was flowing at a rate of 1.2 mL/min, and Cell Gas B was flowing at a rate of 1.4 mL/min. The RPQ value was 0.45;

Comments 4: Line 119: in References item 46 is missing 
Line 154: in References item 47 is missing 
Line 170: in References item 48 is missing 
Line 174: in References item 49 is missing 
Line 185: in References, item 50 is missing
Line 271: in References item 52 is missing 
Line 282: in References item 53 is missing 
Line 285: in References item 54 is missing 
Line 288: in References item 55 is missing 
Line 295: in References item 56 is missing 
Lines 300-301: The mean Ni concentration in soybeans in the current study (9.21 mg/kg) was lower than that in a study in Luxembourg (3.00 mg/kg) [57]. Really?? 
Line 304: in References items 58 and 59 are missing 
Line 308: in References item 60 is missing 
Line 312: in References item 61 is missing 
Line 314: in References item 62 is missing 
Line 318: in References item 63 is missing 
Line 321: in References item 64 is missing 
Line 345: in References item 65 is missing 
Line 357: among the Chinese65??? 
Line 362: than animal fat65,14??? 
Lines 367-368: dietary exposure66??? 
Response 4: We have performed a thorough examination and added all the missing reference.

Comments 5: Lines 370-371: Nickel was detected in food samples, whether nickel concentration was determined in these samples?
Response 5: Thank you for your recommendation. The original expression might be confusing and could potentially lead to misunderstandings, especially regarding the high detection rate of most food samples. We have addressed this concern by revising the sentence in lines 423-424.

Round 2

Reviewer 1 Report

Comments and Suggestions for Authors

Thank you for the Interesting work. However calculating HQ using the average frequency of food intake and nickel concentration in a specific region does not have a significant impact. Instead of using the point estimation like median, 95th, simulation based on the distribution would be recommended. 

Comments on the Quality of English Language

Extensive editing of English language required.

Author Response

Thank you for your insightful idea,  your suggestion would significantly enhance the manuscript. Regarding your recommendation, we have identified that previous study supported using the 90th percentile of metal concentration and the mean of consumption rate in point estimate studies[1]. However, in our current study, the combination of the 90th percentile and mean concentration did not have a significant impact upon assessment. Considering the skewed distribution in the data, we opted for the median and 95th percentile as the point values. Furthermore, instead of relying on single model, we employed four models to represent the HQ distribution by combining the 95th percentile and median. A similar methodology was applied in our previous study, with the reference added in line 239[2].

1.     Gavelek, A.; Spungen, J.; Hoffman-Pennesi, D.; Flannery, B.; Dolan, L.; Dennis, S.; Fitzpatrick, S. Lead Exposures in Older Children (Males and Females 7-17 Years), Women of Childbearing Age (Females 16-49 Years) and Adults (Males and Females 18+ Years):  FDA Total Diet Study 2014-16. Food Addit Contam Part A Chem Anal Control Expo Risk Assess 2020, 37, 104–109, doi:10.1080/19440049.2019.1681595.
2.     Q, W.; R, Z.; P, W.; J, C.; X, P.; D, Z.; J, W.; H, Z.; X, Q.; X, W.; et al. An Occurrence and Exposure Assessment of Paralytic Shellfish Toxins from Shellfish in Zhejiang Province, China. Toxins 2023, 15, doi:10.3390/toxins15110624.

Reviewer 2 Report

Comments and Suggestions for Authors

The authors have thoroughly addressed all the comments and suggestions provided by the reviewers for the manuscript titled "Occurrence and Exposure Assessment of Nickel in Zhejiang Province, China." The revisions have been carefully implemented to reflect the feedback, enhancing the clarity, depth, and scientific rigor of the article. In light of these comprehensive revisions, the manuscript may now be considered acceptable for publication.

Author Response

Thank you for your positive evaluation. We appreciate the constructive feedback provided by the reviewer, and your insightful comments have significantly enhanced the quality of our manuscript. Your guidance has been invaluable.

Reviewer 3 Report

Comments and Suggestions for Authors

The authors have revised the manuscript improving the manuscript

Author Response

Thank you for your evaluation. We sincerely appreciate your constructive feedback, which has significantly enhanced the quality of our manuscript. Your insightful comments are invaluable. Should there be any additional details or further revisions you recommend, please do not hesitate to contact us. We eagerly look forward to receiving any additional feedback you may have. Your expertise is highly valued in improving the overall quality of our manuscript.